# Genome-Edited Fish in the Field

**DOI:** 10.3390/cimb47121013

**Published:** 2025-12-03

**Authors:** Kang Hee Kho, Zahid Parvez Sukhan, Yusin Cho, Doohyun Cho, Cheol Young Choi

**Affiliations:** 1Department of Fisheries Science, Chonnam National University, Yeosu 59626, Republic of Korea; zpsukhan@jnu.ac.kr (Z.P.S.); choys@jnu.ac.kr (Y.C.); chodoo22@jnu.ac.kr (D.C.); 2Division of Marine Bioscience, National Korea Maritime and Ocean University, Busan 49112, Republic of Korea; choic@kmou.ac.kr

**Keywords:** genome editing, aquaculture biotechnology, CRISPR/Cas9, SDN-1, field trial, biosafety, regulatory framework, consumer perception, commercialization

## Abstract

Genome editing using site-directed nucleases (SDNs), particularly with the CRISPR/Cas9 system, has emerged as a powerful platform for aquaculture innovation, enabling precise, heritable, and non-transgenic modifications that enhance productivity, sustainability, and animal welfare. This review synthesizes molecular, regulatory, ecological, and societal perspectives to highlight global advances in genome-edited fish and their transition from laboratory research to field applications. To date, over forty aquatic species have been successfully edited to improve traits such as growth, disease resistance, pigmentation, and reproductive control. Notably, market-approved SDN-1 fish lines, including *mstn*-knockout red seabream and Nile tilapia, and *lepr*-edited tiger puffer and olive flounder, have demonstrated improved productivity; however, publicly available welfare data remain limited. These examples illustrate how product-based SDN-1 regulatory frameworks in Japan, Argentina, and Brazil enable commercialization while ensuring biosafety. Nonetheless, limited field trials and regulatory heterogeneity continue to hinder global harmonization. Major challenges include the development of standardized welfare metrics, assessment of multigenerational stability, evaluation of ecological risks, and transparent data sharing. To address these gaps, a structured reporting checklist is proposed to guide consistent molecular validation, welfare assessment, biosafety containment, and data transparency. Genome editing through SDN-based precision, coupled with ethical governance, represents a crucial step toward sustainable, resilient, and publicly trusted aquaculture systems.

## 1. Introduction

The development of genome editing technologies has opened new opportunities for precision breeding across plants, livestock, and aquaculture. In particular, the Clustered Regularly Interspaced Short Palindromic Repeats and CRISPR-associated protein 9 (CRISPR/Cas9) system has provided researchers with an efficient and versatile tool to introduce targeted genomic changes with unprecedented accuracy. Early demonstrations of CRISPR-mediated mutagenesis in zebrafish (*Danio rerio*) confirmed the feasibility of genome editing in fish [1], and subsequent studies rapidly extended its application to aquaculture species, including Nile tilapia (*Oreochromis niloticus*) [2,3,4], channel catfish (*Ictalurus punctatus*) [5,6], salmonids [7,8], carps [9,10], red seabream (*Pagrus major*) [11], olive flounder (*Paralichthys olivaceus*) [12], and tiger puffer (*Takifugu rubripes*) [13].

Unlike conventional transgenic methods, which typically involve the stable integration of foreign DNA, genome editing enables precise and heritable modifications that mimic naturally occurring mutations. The CRISPR/Cas9 system, an RNA-guided nuclease technology, employs a single-guide RNA (sgRNA) designed from the target locus to direct the Cas9 enzyme to a complementary DNA sequence, where it introduces a double-strand break that is subsequently repaired by the cell through either non-homologous end joining (NHEJ) or homology-directed repair (HDR) [14,15]. This fundamental distinction forms the basis of the site-directed nuclease (SDN) classification framework, which categorizes genome-editing approaches into three main types, SDN-1, SDN-2, and SDN-3, depending on their molecular mechanisms and the presence or absence of exogenous DNA (Figure 1) [16,17,18]. SDN-1 edits create small insertions or deletions at targeted genomic loci through NHEJ without using any repair template or introducing foreign DNA. SDN-2 edits employ short homologous sequences to introduce specific nucleotide substitutions or small sequence changes, resulting in cisgenic outcomes. SDN-3 edits involve the insertion of longer DNA fragments derived from external sources, similar to conventional transgenesis. Most genome-edited (GED) fish developed for aquaculture, including *myostatin* (*mstn*)-knockout red seabream and Nile tilapia, as well as *leptin receptor* (*lepr*)-edited tiger puffer and olive flounder, belong to the SDN-1 category. This classification has significant implications for commercialization and public perception, as SDN-1 organisms are often excluded from genetically modified organism (GMO) regulations in several countries, including Japan, Argentina, and Brazil. Since SDN-1 edits do not introduce foreign DNA, they are generally regarded as equivalent to conventionally bred variants, facilitating streamlined regulatory approval and faster market introduction [18,19]. The SDN framework has thus become a cornerstone of international policy discussions, linking molecular precision with biosafety governance and guiding the responsible development of GED aquaculture.

The aquaculture sector is central to the global food supply, contributing more than half of the aquatic protein consumed by humans and representing the fastest-growing area of food production worldwide [20]. However, this rapid expansion has been accompanied by persistent challenges, including disease outbreaks, fluctuations in feed availability and cost, environmental degradation, and the need to maintain sustainability under increasing climate change pressures. Although traditional selective breeding has achieved notable genetic gains and trait improvements, particularly in salmonids, tilapia, and carps, its progress is constrained by long generation times, complex polygenic traits, and sensitivity to environmental variability [21]. Genome editing has therefore become a complementary approach to overcome these limitations by enabling targeted enhancement of traits such as faster growth, improved feed conversion efficiency, disease resistance, pigmentation modification, and reproductive sterility for environmental containment and market management.

Lessons from the plant sector demonstrate that strong performance under controlled laboratory conditions does not necessarily guarantee similar outcomes in the field. GED crops often behave differently under real-world conditions, where interactions with pathogens, temperature fluctuations, and nutrient availability can influence trait expression in unanticipated ways [22]. The same principle applies to aquaculture, where laboratory-based tank-controlled trials, though essential for molecular validation, cannot fully replicate the dynamic complexity of ponds, net cages, or open-water systems. Consequently, field and semi-field evaluations are indispensable for determining whether GED fish deliver the intended benefits under commercial and ecological conditions, as well as for identifying any unforeseen trade-offs. Importantly, GED fish are no longer merely hypothetical. Several species have already advanced to commercialization, marking a milestone in aquaculture biotechnology. However, the trajectory of genome editing in aquaculture mirrors that of plant biotechnology worldwide, characterized by rapid laboratory adoption followed by divergent regulatory interpretations. In the European Union (EU), the European Court of Justice ruled in 2018 (Case C-528/16) that organisms developed through new mutagenesis techniques are subject to the same stringent regulations as GMOs. Because GED organisms fall under the same legal framework as GMOs in the EU, commercial authorization is highly stringent, and although experimental releases are possible under Part B authorization, field trials remain limited compared to countries operating product-based SDN-1 frameworks [23]. In contrast, countries such as Japan, Argentina, Brazil, and several others have adopted product-based or case-by-case regulatory frameworks that exempt SDN-1 edits from GMO legislation, provided that no foreign DNA remains in the final product (Figure 2) [17,24]. Under these systems, GED organisms are evaluated based on the characteristics of the final product rather than the process used to produce them.

This regulatory divergence has tangible implications for the research and commercialization of GED fish. In Japan, SDN-1 GED red seabream with disrupted *mstn* for enhanced muscle growth, tiger puffer with edited *lepr* for appetite regulation, and olive flounder with similar appetite-related modifications have been approved for food use and sale, subject to mandatory labeling and transparent consumer communication [18]. These approvals were made possible precisely because the products were classified as SDN-1 edits without foreign DNA, exempting them from transgenic regulations while still requiring biosafety self-assessment. Beyond Asia, Argentina and Brazil have authorized the production of SDN-1 *mstn*-edited Nile tilapia, adopting product-based regulatory systems that evaluate the characteristics of the final organism rather than the process used to create it [25]. Together, these examples highlight how the SDN classification has become central to the governance of GED aquaculture, supporting science-based differentiation, regulatory clarity, and consumer transparency.

Evaluating GED fish in farming environments presents unique challenges that differ from those encountered in plant or terrestrial livestock systems. Aquatic environments are inherently porous, making complete containment difficult and raising concerns about escape and gene flow into wild populations [26]. Escapes are already a well-known issue in conventional aquaculture, where escaped individuals can interbreed with wild conspecifics or disrupt ecological interactions [26,27]. For GED fish species, potential ecological consequences include competition with wild individuals, altered trophic dynamics, and introgression of edited alleles into natural populations. Addressing these risks requires multidisciplinary data on edit durability across generations, mosaicism and off-target effects in founder lines, welfare outcomes at commercial densities, trait stability across temperature and salinity gradients, the efficacy of sterility systems, and traceability mechanisms that maintain consumer trust and regulatory oversight [28].

This review is presented at a decisive moment for aquaculture biotechnology. The commercialization of SDN-1 GED fish has already begun in multiple jurisdictions, yet empirical data on their field and semi-field performance remain fragmented and often inaccessible across studies, languages, and institutional boundaries. At the same time, regulatory frameworks continue to diverge internationally, creating uncertainty for researchers, producers, and policymakers. A unified vocabulary that links molecular categories such as SDN-1, SDN-2, and SDN-3 to measurable risk and benefit outcomes is urgently needed. Furthermore, decisions made in the present decade regarding investment, biosafety assessment, and labeling will shape the future trajectory of GED aquaculture, including the development of traits for climate resilience, disease resistance, and welfare optimization. Accordingly, this review consolidates current knowledge on GED fish development, with an emphasis on field-relevant performance, biosafety considerations, and societal contexts. By integrating biological, environmental, and policy perspectives within the SDN regulatory framework, this article aims to provide an evidence-based foundation for responsible innovation, transparent governance, and long-term sustainability as GED fish transition from laboratory systems to commercial reality.

## 2. Current Status of GED Fish Development and Field Trials

The application of genome editing in aquaculture has expanded rapidly over the past decade, progressing from experimental proof-of-concept studies to the early phases of field validation and commercialization. Since the first documentation of CRISPR/Cas9-mediated mutagenesis in zebrafish in 2013 [1], researchers have successfully applied this technology across a wide range of economically important finfish and shellfish species [28,29]. Research efforts have gradually shifted from exploratory demonstrations to the targeted improvement of production traits, supported by advances in editing precision, germline transmission, and regulatory clarity. Major objectives include accelerating growth, enhancing feed conversion efficiency, improving disease resistance, controlling reproduction, altering pigmentation, reducing intermuscular bones (IBs), and enhancing welfare-related traits such as reduced aggression or calmer swimming behavior. In parallel, progress in SDN classification, optimization of molecular tools, and the initiation of farm-scale evaluations have accelerated the transition of laboratory findings into commercial applications [18,29,30]. Collectively, these developments have positioned aquaculture as one of the most dynamic frontiers of innovation in animal biotechnology.

### 2.1. Expansion of Genome Editing Applications in Aquaculture

Since the first successful demonstration of CRISPR/Cas9 genome editing in zebrafish, the application of this technology in aquaculture species has expanded rapidly. As of 2025, successful edits have been reported in more than forty fish and shellfish species, encompassing both exploratory and production-oriented research (Table 1). Major aquaculture taxa that have undergone genome editing include salmonids, tilapia, catfish, carps, red seabream, olive flounder, tiger puffer, and bluefin tuna. While zebrafish and medaka continue to serve as foundational models for tool development and proof-of-concept validation, applied research has increasingly focused on high-value aquaculture species in which genome editing can directly improve growth performance, feed efficiency, disease resistance, and other key production traits.

The scope of genome editing in aquaculture has expanded beyond single-gene mutagenesis to induce multiplex CRISPR systems that enable the simultaneous disruption or modification of multiple loci. Innovations such as base editors and prime editors now allow precise single nucleotide substitutions without inducing double-strand breaks [67,68]. These technical advances have reduced mosaicism and off-target effects in founder generations while improving the heritability of edited alleles. In parallel, progress in germline editing, surrogate broodstock technology, and gamete cryopreservation has further facilitated the stable propagation and dissemination of GED lines within selective breeding programs [30,69].

The integration of CRISPR-based approaches with genomic selection has accelerated the translation of genetic discoveries into practical applications. Enhanced genomic resources, coupled with clearer regulatory frameworks, are now supporting the transition of GED fish from laboratory systems to semi-field and commercial environments [18,24,70]. Collectively, these advances position aquaculture as one of the leading animal production sectors adopting genome editing for sustainable growth and food security.

### 2.2. Trait Categories and Target Genes

Genome editing in aquaculture has diversified rapidly to target traits directly related to productivity, welfare, and market value. The most intensively pursued traits include muscle growth regulation, appetite control and feed efficiency, reproductive control, skeletal morphology, pigmentation, disease resistance, and behavior. Disruption of *mstn*, a negative regulator of muscle development, has been reported in several species including red seabream, olive flounder, pufferfish, Nile tilapia, and common carp, resulting in 15–30% greater muscle mass without external deformities (Figure 3A) [11,12,13,71,72]. Editing of *lepr* gene has been used to enhance appetite and feed efficiency in tiger puffer and olive flounder [73]. Reproductive control genes such as *follicle-stimulating hormone beta subunit* (*fshb*), *luteinizing hormone beta subunit* (*lhb*), or *dead end homolog* (*dnd*) have been targeted to induce sterility and minimize the risk of gene flow, thereby strengthening biosafety in open systems (Figure 3B) [36,41,60].

Genome editing has also been employed to enhance product quality in aquaculture species. In carps and other bony fishes, targeted editing of the *bone morphogenetic protein 6* (*bmp6*) and *runt-related transcription factor 2b* (*runx2b*) genes has been shown to reduce or eliminate IBs, thereby improving filet yield and processing efficiency without compromising growth performance. These skeletal-targeted edits illustrate how genome editing can improve consumer acceptance while maintaining production efficiency. Editing of the *bmp6* gene in silver carp (*Hypophthalmichthys molitrix*) resulted in approximately a 30% reduction in IBs in the F0 generation [64]. Similarly, disruption of *bmp6* in crucian carp (*Carassius auratus cuvieri*) produced a 40.74% reduction in IBs in the F0 generation and complete elimination by the F3 generation [55]. Furthermore, editing of the *runx2b* gene in Gibel carp (*Carassius gibelio*) led to partial elimination of IBs in the F0 generation and complete elimination in the F1 generation (Figure 4) [56].

Behavioral and physiological traits have also been identified as strategic targets for enhancing welfare and energy efficiency in aquaculture species. In Pacific bluefin tuna (*Thunnus orientalis*), targeted editing of the *ryanodine receptor 1b (RyR1b)* gene alters calcium signaling in muscle cells, leading to slower swimming behavior and lower energy expenditure, thereby improving handling under aquaculture conditions [45]. Similarly, disruption of the *arginine vasotocin receptor v1a2* (*v1a2*) gene in chub mackerel (*Scomber japonicus*) significantly reduced aggressive behavior in fry, mitigating fin damage and mortality under high-density rearing [76].

Pigmentation modification has become a distinctive application of genome editing, serving both functional and commercial purposes. Knockout of melanin synthesis genes such as *tyrosinase* (*tyr*), *solute carrier family 45 member 2* (*slc45a2*), *premelanosome protein* (*pmel*), *oculocutaneous albinism II* (*oca2*), and *microphthalmia-associated transcription factor a* (*mitfa*) has produced albino, golden, or transparent strains in mackerel tuna (*Euthynnus affinis*), Malawi cichlid (*Astatotilapia calliptera*), Nile tilapia, and carps [49,51,77,78]. In ornamental and market species, such pigmentation control enables visual distinction of GED lines and enhances consumer appeal. For instance, disruption of *tyr* in fathead minnow (*Pimephales promelas*) and *oca2* in Malawi cichlid yielded pale or golden phenotypes that improve visibility during culture and provide a clear identity for commercial distribution [47,51]. Moreover, pigmentation edits offer a noninvasive phenotypic marker for confirming edit heritability without molecular assays, providing practical advantages in breeding and traceability systems.

Disease resistance is another expanding frontier in genome editing for aquaculture. In Atlantic salmon (*Salmo salar*), editing of the *neddylation activating enzyme 1* (*nae1*) gene enhanced resistance to infectious pancreatic necrosis virus [79]. In zebrafish, disruption of *forkhead box O3b* (*foxo3b*) and *fms-related tyrosine kinase receptor 4* (*flt4*) enhanced innate immune responses [80,81]. In grass carp (*Ctenopharyngodon idella*), knockout of *junctional adhesion molecule a* (*jam-a*) and *phosphatidylinositol 4-kinase beta* (*pi4kb*) genes reduced replication of grass carp reovirus (GCRV) and improved growth performance [39,82]. Channel catfish (*Ictalurus punctatus*) has also served as a platform for an innovative dual-trait approach combining disease resistance and reproductive control. By inserting an alligator-derived *cathelicidin* (*cath*) gene at the luteinizing hormone (*lh*) locus, researchers produced fish with enhanced resistance to bacterial pathogens while simultaneously achieving sterility, demonstrating how trait stacking can address both aquaculture performance and biosafety [83]. Collectively, these examples illustrate how genome editing in fish has progressed from exploratory mutagenesis to trait-driven innovation aimed at improving productivity and consumer appeal, with some traits also proposed as having potential welfare benefits; however, empirical welfare assessments remain limited.

### 2.3. Transition from Laboratory to Farm-Scale Evaluation

The progression of GED fish from controlled laboratory settings to field-scale production marks a pivotal phase in aquaculture biotechnology. Most early CRISPR/Cas9 studies were conducted under laboratory or hatchery conditions, primarily to confirm editing efficiency, heritability, and phenotypic stability. However, translating these traits into commercial environments requires testing under variable ecological and management conditions that reflect real-world aquaculture systems [18,29].

Japan has taken the global lead in implementing this transition. Under the product-based case-by-case notification system jointly administered by the Ministry of Agriculture, Forestry and Fisheries (MAFF) and the Ministry of Health, Labour and Welfare (MHLW), three GED fish lines have achieved market entry: *mstn*-edited red seabream, *lepr*-edited tiger puffer, and olive flounder. These fish were developed collaboratively by Kyoto University and the Regional Fish Institute using DNA-free CRISPR/Cas9 methods that induce small indels through NHEJ without introducing foreign DNA [84]. Multi-year grow-out trials in land-based aquaculture systems demonstrated substantial improvements in growth rate and feed conversion efficiency, with no adverse health or morphological effects. These data provided the empirical basis for the first approval of GED fish in Japan, which are sold with voluntary labeling that explicitly discloses the genome editing process [17,18,85,86].

Latin America has become the second region to advance GED aquaculture towards production-scale application. In Argentina and Brazil, *mstn*-edited Nile tilapia were jointly developed by U.S. biotechnology companies and regional partner institutes [29,87,88]. Editing was achieved via CRISPR/Cas9-induced frameshift mutations consistent with the SDN-1 classification, resulting in higher growth rate and improved filet quality compared to unedited counterparts [89]. Regulatory determinations by the National Advisory Commission on Agricultural Biotechnology (CONABIA) in Argentina and the National Technical Biosafety Commission (CTNBio) in Brazil concluded that SDN-1 organisms lacking foreign DNA do not fall within their GMO frameworks [90,91]. Consequently, pilot commercial production began in 2025 in Brazil under product-based regulatory oversight, making the first international example of GED fish commercialization beyond Japan [92].

Together, these pioneering cases demonstrate that GED fish can be safely reared under commercial conditions while maintaining welfare and biosafety standards. The transparent labeling system in Japan and the product-based regulatory frameworks in Latin America offer two complementary models for the responsible market introduction of GED fish. While these achievements confirm the feasibility of genome editing for aquaculture production, large-scale validation across diverse environments and species remains limited. The following section outlines emerging field efforts and research priorities needed to strengthen the evidence base for GED fish.

### 2.4. Emerging Field Trials and Research Gaps

The transition from controlled research to farm-scale application remains one of the most challenging steps in GED fish development. Although laboratory studies have established strong molecular and phenotypic foundations, only a limited number of species have progressed to field evaluation under realistic aquaculture conditions [29,84]. Most ongoing projects remain limited to recirculating aquaculture systems or indoor tanks, which limit exposure to environmental variability. In contrast, field environments introduce fluctuating temperature, salinity, microbial interactions, and stocking densities, all of which may influence gene expression, growth, and stress responses [70,93].

Beyond Japan and Latin America, several countries are initiating pilot-scale GED fish assessments. In China, a food safety and environmental impact evaluation was completed for an all-female GED common carp line, representing a crucial milestone toward pre-commercial testing [94]. Programs in the United States and Norway are exploring CRISPR applications in salmonids for growth and disease resistance, while early-stage research in Southeast Asia focuses on catfish and carps aligned with regional aquaculture priorities [29,70]. Although promising, these initiatives are not yet standardized in terms of experimental design, welfare assessment, or environmental monitoring, making comparisons across regional challenging.

The four existing GED fish lines (seabream, tiger puffer, olive flounder, and Nile tilapia) represent the first generation of GED aquatic animals entering aquaculture markets. These species share several defining features: (1) single-gene SDN-1 edits targeting growth- or appetite-related pathways, (2) DNA-free delivery methods that simplify regulatory approval, and (3) product-based oversight focusing on the final phenotype rather than the editing process. Although regulatory frameworks differ among jurisdictions, a common success factor has been regulatory clarity paired with transparent consumer communication. These pioneering examples confirm the technical feasibility and social manageability of GED fish while also revealing critical gaps, including the limited availability of peer-reviewed farm-scale performance data, long-term ecological risk assessments, and standardized welfare metrics (Table 2). Addressing these gaps will be essential to ensure that future GED aquaculture expands responsibly, delivering both productivity gains and public trust.

Across these national contexts, several recurring research and methodological gaps continue to limit synthesis and risk assessment:(1)*Standardization of evaluation protocols*: There is currently no unified framework for assessing performance, welfare, and ecological parameters in GED fish trials [28,95].(2)*Multi-generational assessment*: Most published studies report results only up to the F1 or F2 generations, with few tracking lineages over multiple generations to evaluate heritability, edit stability, and potential long-term fitness trade-offs [96].(3)*Ecological interaction studies*: Quantitative data on behavior, competition, reproductive success in mixed populations, and ecological fitness in open systems remain scarce. The ecological implications of potential escapes or gene flow under real-world conditions are largely unexplored in the context of GED fish [29,97].(4)*Public transparency and data accessibility*: Many genome editing projects remain documented only in internal reports, regulatory dossiers, or industry announcements rather than peer-reviewed publications. This restricts opportunities for independent validation, replication, and accumulation of scientific knowledge [98].

Addressing these gaps will require coordinated international collaboration, open-access sharing of field data, and harmonized reporting standards that integrate performance, welfare, and biosafety metrics. Lessons from Japan demonstrate that transparent governance, traceable labeling, and proactive public communication are essential for prompting public trust as GED fish transition from research facilities to commercial aquaculture systems.

## 3. GED Fishes in the Field

The transition of GED fish from research facilities to commercial aquaculture represents a pivotal milestone in applying CRISPR/Cas technology to food production. Following a decade of laboratory validation, several GED lines have advanced to real-world farming and market stages. These include the *mstn*-knockout red seabream, *lepr*-edited tiger puffer and olive flounder developed in Japan, as well as *mstn*-modified Nile tilapia commercialized in Argentina and Brazil (Table 3). Collectively, these examples demonstrate how well-defined regulatory frameworks, product-based oversight, and transparent labeling can accelerate the responsible adoption of GED organisms in aquaculture. However, the extent and scope of these regulatory assessments remain difficult to evaluate because most underlying reports are not publicly accessible, and it is unclear whether systematic welfare or health evaluations were included beyond food safety considerations.

### 3.1. Genome Edited Fish in the Field in Japan

#### 3.1.1. Unified Regulatory Pathway of SDN-1 GED Fish in Japan

Japan applies a product-based SDN-1 regulatory framework in which GED organisms without foreign DNA insertion fall outside the scope of the Cartagena Act on GMOs [99,100,101]. Developers must notify both the Ministry of Agriculture, Forestry and Fisheries (MAFF) and the Ministry of Health, Labour and Welfare (MHLW), providing evidence of the target gene, editing method, mutation type, sequencing validation, and confirmation of the absence of any vector-derived sequences [18,84,102,103,104]. After notification, the Biosafety Impact Assessment Committee evaluates ecological risk and determines containment and monitoring requirements based on species, facility architecture, and production scale [105,106,107]. Commercial release is accompanied by a voluntary transparency-based labeling system in which GED foods are clearly identified and traceability information is made accessible through QR codes [18,84]. This unified procedure applies to all commercialized GED aquatic organisms in Japan and serves as regulatory template for future species. To date, three GED fish lines have been commercialized under this framework.

#### 3.1.2. Red Seabream

The GED red seabream represents the first officially approved GED animal in Japan to enter the food market and a landmark achievement in aquaculture biotechnology. The strain, developed through a collaboration between Kyoto University and the Regional Fish Institute, was produced by disrupting the *mstn* gene, which encodes a negative regulator of muscle development [11,83]. Using CRISPR/Cas9 technology, researchers introduced a 14 bp deletion in exon 1 of *mstn*, generating a loss-of-function mutation that suppressed myostatin expression and promoted muscle hypertrophy [11,108]. As a result, the edited fish exhibited approximately 16–18% greater body weight, a 20% increase in filet yield, and 6–14% enhancement in feed conversion efficiency compared with the conventional strain (Figure 5). In addition, the edited fish displayed 9% greater body width and a 7% increase in body height corresponding to visibly thicker muscle sections while maintaining normal skeletal conformation [11].

Genomic validation included whole-genome sequencing of F2 progeny to confirm the absence of residual Cas9 mRNA, sgRNA fragments, or plasmid backbone, and no off-target edits were identified among genomic loci with up to two mismatches [102,108]. Multi-generational breeding showed stable inheritance and normal reproductive performance, including fecundity, fertilization rate, egg viability, and hatchability [108]. Commercial rollout began in 2021 through direct-to-consumer online sales and distribution to certified restaurants. Initial market introduction emphasized traceability and transparency, and demographically stratified surveys indicated moderate-to-high acceptance among consumers when information on editing purpose, traits, and absence of foreign DNA was provided [84,85].

The case of the GED red seabream in Japan establishes a precedent for the safe and transparent commercialization of SDN-1 fish. It demonstrates that DNA-free CRISPR/Cas9 editing can produce heritable and commercially desirable phenotypes without introducing foreign genes, while meeting rigorous standards for biosafety and consumer communication. This example has since guided national policy development on GED aquatic organisms and serves as a reference model for similar initiatives in other countries.

#### 3.1.3. Tiger Puffer

The tiger puffer, known as *torafugu* in Japan, represents the second GED fish line to enter the commercial food market, following the GED red seabream. A research team at Kyoto University and the Regional Fish Institute employed the CRISPR/Cas9 platform to disrupt the *lepr gene*, which plays a key role in regulating appetite and energy metabolism [110]. The knockout of *lepr* results in a loss of leptin signaling, which suppresses satiety, thereby increasing feed intake and enhancing growth performance. The edited line was officially designated “*high-growth tiger puffer* (*4D-4D strain*)” under Japan’s MHLW notification system.

The F1 generation exhibited a 4 bp deletion in *lepr*, resulting in a nonfunctional receptor. Feeding trials in land-based rearing facilities demonstrated that the edited fish showed markedly improved feed conversion efficiency and accelerated weight gain, achieving approximately 1.5- to 1.9-fold higher growth compared with unedited controls under identical feeding conditions (Figure 6) [100,103]. Histological and physiological evaluations revealed no abnormalities in gonadal development, hepatopancreas structure, or overall health, and spawning traits remained within the range observed in control groups. Based on the verified absence of foreign DNA and the lack of identified environmental risk, the GED tiger puffer was classified as an SDN-1 organism and thereby exempted from transgenic regulation under Japan’s Food Sanitation Act. In 2021, MAFF and MHLW approved the commercial distribution of GED tiger puffer, making it the second GED animal cleared for food use in Japan [18,84].

The tiger puffer case underscores the practical effectiveness of Japan’s notification-based SDN-1 framework, demonstrating that GED fish can progress from laboratory validation to consumer markets through a transparent and science-based regulatory pathway. It also highlighted that land-based, closed-system aquaculture, combined with rigorous containment measures and proactive disclosure, can provide a socially accepted model for the safe commercialization of GED fish.

#### 3.1.4. Olive Flounder

The olive flounder, one of Japan’s most important flatfish aquaculture species, became the third GED fish line to receive official authorization for food use in 2023. The GED line, designated as the “8D strain” was developed by the Regional Fish Institute in collaboration with Kyoto University. This strain was produced using the CRISPR/Cas9 system to disrupt the *lepr* gene, which plays a central role in appetite regulation and energy balance [112]. The editing strategy followed the same leptin-signaling mechanism previously applied to the tiger puffer, aiming to enhance feeding activity and growth performance in a benthic species adapted to low-energy habitats.

Under controlled aquaculture conditions, the GED olive flounder exhibited approximately 20–25% greater average harvest weight relative to the unedited control strain, enabling earlier attainment of market size [101]. No abnormalities were observed in morphology, reproductive characteristics, or health indices such as hepatopancreatic and gonadal histology [101]. The edited strain also maintained normal behavior and survivability throughout the rearing period, supporting the interpretation that appetite stimulation did not impose negative physiological trade-offs. Commercial distribution started in 2024 through online retailers and regional seafood channels, with voluntary genome-editing labeling, which emphasized DNA-free editing and the SDN-1 classification [113].

The case of the olive flounder highlights Japan’s stepwise and transparent approach to regulating GED fish. By combining molecular precision with rigorous containment and disclosure practices, this initiative expands the diversity of GED species in commercial production and sets a model for the sustainable implementation of aquaculture biotechnology under public oversight.

### 3.2. Nile Tilapia in South America

Nile tilapia was the first aquaculture species outside Japan to receive regulatory determinations permitting the commercialization of a GED line. In Argentina, the National Advisory Commission on Agricultural Biotechnology (CONABIA) issued a case-by-case determination in late 2018, confirmed in early 2019, concluding that a CRISPR/Cas9-edited Nile tilapia line targeting the *mstn* gene to enhance growth and filet yield does not fall under national GMO regulations because it contains no new combination of genetic material. The line, designated FLT-01, was developed by the U.S.-based biotechnology company AquaBounty, a subsidiary of Intrexon Inc. [114]. Genome editing was performed by microinjecting Cas9 nuclease and sgRNAs into the one-cell embryos to induce small indels at the *mstn* locus via NHEJ. This process generated frameshift mutations, producing loss-of-function alleles consistent with the SDN-1 classification. The edited FLT-01 line exhibited approximately a 70% improvement in filet yield, a 16% increase in growth rate, and a 14% enhancement in feed conversion ratio compared with wild-type controls [87,89,114]. The DNA-free lines were subsequently assessed by CONABIA under Argentina’s product-based biosafety framework, which distinguishes SDN-1 organisms from transgenic GMOs.

In Brazil, the National Technical Biosafety Commission (CTNBio) regulates new breeding technologies under Normative Resolution No. 16/2018 (NR-16), which provides a case-by-case mechanism to determine whether such products qualify as non-GMOs when no foreign DNA is integrated [90,115]. Utilizing this product-based framework, the Brazilian company “Brazilian Fish” announced in 2025 a commercial-scale GED Nile tilapia program developed in partnership with the US-based Center for Aquaculture Technologies (CAT). The initiative aimed to improve growth and filet yield through CRISPR-based SDN-1 editing, aligning commercialization with Brazil’s non-GMO classification under NR-16 [92,116]. Public communications suggest that the Brazilian line was independently developed for domestic production rather than derived directly from Argentina’s FLT-01 strain, although both share the same SDN-1 foundation and product-based regulatory principles.

Together, these developments illustrate how SDN classification and product-based, case-by-case review have enabled the early market entry of GED Nile tilapia while maintaining biosafety oversight. The Argentine determination of FLT-01 established an international precedent by explicitly recognizing the absence of new genetic combinations in SDN-1 animals, while Brazil’s NR-16 provides a clear legal pathway for domestic producers to commercialize GED fish when no foreign DNA is present [29,90,115]. In both jurisdictions, such products are exempt from special GMO labeling when edits meet SDN-1 criteria, contrasting with Japan’s mandatory labeling approach for GED food. As farm-scale data accumulate, the Nile tilapia case will continue to shape both regulatory discourse and the practical integration of genome editing into sustainable freshwater aquaculture systems.

## 4. Regulatory and Policy Landscape of GED Fish

The governance of GED fish represents one of the most dynamic and unsettled areas in contemporary biotechnology policy. Although the technical distinction between genome editing and transgenesis is now broadly recognized, regulatory interpretations differ substantially across jurisdictions in terms of risk assessment, labeling, and market approval. Some countries, such as Japan, Argentina, and Brazil, have adopted product-based regulatory frameworks that evaluate the characteristics and safety of the final organism rather than the process used to produce it. In contrast, Canada applies a product-oriented regulatory trigger based on trait novelty, whereas the EU applies a combined trigger in which the mutagenesis technique determines inclusion under the GMO framework and the resulting organism is then assessed on a case-by-case basis. Across both paradigms, most authorities emphasize case-by-case reviews, where oversight is proportionate to the type of edit, ecological context, and intended use rather than imposed through uniform classifications. The diversity of these approaches reflects not only scientific differences but also political, ethical, and societal priorities concerning food technology, environmental protection, and consumer transparency [18,24,30,117]. As GED fish move toward commercialization, understanding this evolving regulatory mosaic is crucial for enabling international harmonization, maintaining consumer trust, and supporting safe innovation in aquaculture.

### 4.1. Diverging Global Approaches

Globally, two primary regulatory paradigms dominate biotechnology governance. Systems with process-triggered GMO classification, typified by the EU, focus on the mutagenesis techniques used to modify organisms, after which case-by-case product evaluation follows, whereas product-based systems, as implemented in Japan and several Latin American countries, emphasize the traits and risk profile of the final organism [29,118].

In Japan, the Ministry of Environment (MOE) and the Ministry of Health, Labor, and Welfare (MHLW) determined in 2019 that organisms edited without the insertion of foreign DNA fall outside the scope of the Cartagena Act on biosafety [119]. Each product must be individually notified and evaluated on a case-by-case basis, providing details about the editing method, target gene, and verification that no foreign DNA remains [120]. In the United States, the Department of Agriculture (USDA) exempts GED animals when the genetic change could occur through traditional breeding, while the Food and Drug Administration (FDA) retains case-specific oversight for edits that may affect animal health, food composition, or safety [121,122]. In contrast, the EU regulates all GED organisms as GMOs under Directive 2001/18/EC following the 2018 European Court of Justice ruling (Case C-525/16), which classified SDN products as GMOs. However, the current EU regulatory proposal on new genomic techniques applies exclusively to plants; GED animals are not included in this reform and will continue to be regulated as GMOs under Directive 2001/18/EC [123].

### 4.2. Asia-Pacific Leadership: Japan and China

Japan was the first country to approve GED fish for human consumption. Under the 2019 MHLW-MAFF notification system, developers must disclose the editing methods, target genes, molecular verification that no foreign DNA is present, and results of phenotypic evaluations. Once confirmed as SDN-1, products are publicly listed for marketing and sold with mandatory labeling as “genome-edited organisms”. As of 2025, three species, red seabream (*mstn*-knockout), tiger puffer (*lepr*-edited), and olive flounder (*lepr*-edited), have been commercialized under this system [18,84]. Transparent labeling, consumer education, and public engagement have been credited with maintaining high levels of consumer confidence [84]. In China, the Ministry of Agriculture and Rural Affairs (MARA) introduced pilot guidelines in 2022 for field testing and safety evaluation of GED crops and livestock. Although no aquatic species have yet received commercial approval, several universities and research institutes have initiated trials targeting various economically important traits and biosafety assessments [84,94,124].

### 4.3. The Americas: Product-Based Oversight

In the Americas, product-based evaluation is the dominant regulatory approach. Argentina’s Resolution No. 173/2015 assesses whether a new organism contains a “new combination of genetic material”. If no foreign DNA is introduced, the product may be excluded from GMO regulation. Using this pathway, *mstn*-edited Nile tilapia (FLT-01) was exempted from GMO classification by CONABIA in 2019 [29,125]. Similarly, under Normative Resolution No. 16/2018 in Brazil, CTNBio applies the same principle, granting case-by-case exemptions for GED organisms that do not contain recombinant DNA. Publicly listed determinations include *mstn*-edited Nile tilapia currently entering pilot commercialization [92]. Both Argentina and Brazil maintain documentation and biosafety evaluations but focus risk assessment on the traits of the final product rather than the genome-editing method.

### 4.4. Labeling and Traceability Practices

Labeling and traceability are critical for maintaining consumer confidence in GED foods. Japan mandates explicit labeling for all GED products and requires sellers to provide accessible information explaining the nature and purpose of the edit, as well as its distinction from transgenic modification (i.e., the absence of foreign DNA). QR code-linked traceability allows consumers to verify product information through the official databases of the MHLW and MAFF [99]. By contrast, Argentina and Brazil do not require specific labeling for SDN-1 organisms classified as non-GMOs, in line with their product-based regulatory philosophy. Within the EU, however, GED animals remain subject to full GMO labeling under Regulation (EC) No. 1829/2003. Detection and traceability remain technical challenges, as SDN-1 edits are often indistinguishable from naturally occurring mutations [70]. To support international harmonization and trade, the Codex Alimentarius Commission and the OECD Working Group for the Harmonization of Regulatory Oversight in Biotechnology are developing documentation-based traceability standards [70,126]. The diversity of national frameworks and SDN classifications is summarized in Table 4, which compares major regulatory systems, responsible agencies, labeling requirements, and currently authorized GED fish species.

### 4.5. Ethical, Societal, and Trade Implications

Divergent national policies risk creating non-tariff barriers for aquaculture products. Fish approved under product-based regulatory systems may encounter import restrictions in countries that classify all GED organisms as GMOs. Therefore, harmonization and mutual recognition mechanisms are essential to prevent trade asymmetries and regulatory conflicts [24]. Ethical and societal considerations include the ownership of edited lines, equitable access to biotechnology, and ensuring informed consumer choice through transparent labeling. Surveys indicate that clear communication regarding the purpose of genome edits, particularly when associated with sustainability or animal welfare, can enhance public acceptance [18].

Overall, global governance of GED fish remains fragmented. Japan, Argentina, and Brazil have established transparent, product-based pathways that facilitate market entry while maintaining biosafety oversight. In contrast, the EU and Canada continue to apply restrictive or precautionary frameworks. Developing internationally recognized standards for documentation, labeling, and biosafety evaluation will be crucial to ensure safety, transparency, and societal legitimacy as GED fish become integrated into aquaculture markets.

## 5. Consumer Perception and Communication

Public acceptance remains one of the most critical bottlenecks in the adoption of GED fish. Even when technical performance and biosafety are well established, consumer trust and social legitimacy ultimately determine market success. In practice, public attitudes toward GED aquaculture are shaped by awareness, institutional trust, perceived naturalness, and cultural framing. Understanding these dimensions is therefore essential to ensure that genome editing contributes to sustainable food systems rather than triggering renewed controversy.

### 5.1. Public Awareness and Knowledge Gaps

Studies consistently show that consumers often conflate genome editing with traditional genetic modification, leading to confusion and skepticism in public discourse. Surveys conducted in Japan, Europe, and the Americas indicate that while many people recognize the term “GM” or “genetic modification”, relatively few understand the nuances of CRISPR-mediated edits or can reliably distinguish between transgenesis and gene editing [17,18]. As a result, public perceptions of GED foods are frequently inherited from earlier GMO controversies rather than informed by evidence-based assessments of risk or benefit.

In Japan, awareness of genome editing was initially low, but proactive outreach by government, research institutes, and industry has gradually improved understanding and acceptance [18]. A recent longitudinal study in Japan found a positive correlation between familiarity and acceptance: as citizens learn how genome editing differs from gene transfer, support for its application in aquaculture and agriculture increases [127]. In contrast, in regions with limited exposure to aquaculture or biotechnology (some parts of Europe and inland areas), baseline awareness is even lower, and skepticism tends to dominate [128].

Trust in institutions also correlates strongly with acceptance. Public surveys of agricultural CRISPR applications show that respondents who trust regulatory agencies, scientists, or public health institutions tend to perceive lower risk and demonstrate greater acceptance of GED products. Conversely, suspicion of hidden agendas or opaque product development processes can amplify perceived risks beyond what is supported by the scientific evidence [128,129].

### 5.2. Factors Influencing Acceptance

Empirical studies identify three interrelated drivers of consumer acceptance of GED products: perceived naturalness, transparency, and perceived benefits.

#### 5.2.1. Perceived Naturalness

Edits that mimic naturally occurring mutations, avoid insertion of foreign genes, or result in minor allelic changes are generally judged more acceptable than those introducing exotic traits or transgenes. In public attitude experiments, participants rated “genome edited” more favorably when it was framed as “precision breeding” or “DNA editing” rather than “genetic engineering”. This framing effect is amplified when the edit addresses health or ecological concerns [129]. Reluctance tends to increase for more invasive edits perceived as “unnatural” [130].

#### 5.2.2. Transparency

Transparent labeling, traceability, and public access to safety data are decisive for building consumer confidence. Studies show that clear explanations of editing methods, target genes, and safety evaluations consistently increase willingness to accept GED products [131]. In Japan, government-mandated disclosure and the use of QR codes on GED fish packaging, which link to official MHLW databases, have been widely credited with maintaining consumer trust [18,84]. Recent consumer studies support these observations. Studies in Japan and Europe indicate that consumer acceptance of genome-edited foods increases when labeling and traceability information are provided, especially when editing purpose and absence of foreign DNA are clearly communicated [17,128,130]. Transparent disclosure has also been shown to reduce perceived risks and increase willingness to purchase genome-edited seafood in both online and restaurant markets [17,86,128,131]. Moreover, research indicates that presenting genome editing as a tool to reduce antibiotic use or improve sustainability significantly lowers opposition.

#### 5.2.3. Perceived Benefits

Perceived benefit is the strongest single predictor of acceptance. Consumers are more likely to support GED products when they perceive direct personal or public advantages, such as enhanced nutritional value, environmental sustainability, or reduced chemical use, rather than purely economic benefit for producers [129,131,132]. Cross-cultural comparisons reveal that emphasizing institutional transparency and food safety oversight promotes relatively stable acceptance levels in Japan, whereas European discussions tend to foreground ethical and ecological considerations, moderating overall support [133].

### 5.3. Communication Strategies

For GED aquaculture to achieve social legitimacy, communication strategies must evolve from one-way information dissemination to interactive and participatory engagement, thereby promoting transparency, trust, and understanding.

#### 5.3.1. Multi-Channel Transparency

Public databases should be established to describe editing methods, target genes, and safety assessment outcomes. These should be complemented by QR codes on product packaging to ensure full traceability and provide consumers with direct access to verified information at the point of purchase.

#### 5.3.2. Purpose Framing

Communication should emphasize edits that contribute to sustainability, animal welfare, and reduced chemical use rather than focusing solely on production efficiency or economic gain.

#### 5.3.3. Intermediary Empowerment

Collaboration with chefs, retailers, fisheries cooperatives, and trusted local stakeholders can help contextualize GED fish as responsibly bred and transparently regulated products, rather than as abstract products of industrial laboratory outcomes.

#### 5.3.4. Responsive Labeling

A two-tier labeling system can enhance understanding and trust. This may combine a concise mandatory statement (for example, “genome-edited for improved growth efficiency”) with optional digital links that provide detailed information about the edit, safety assessments, and regulatory status.

#### 5.3.5. Consistent and Carefully Chosen Language

Neutral, descriptive terms such as “gene editing” or “precision editing” should be used, while emotionally charged or misleading expressions like “genetic engineering” or “engineered fish” should be avoided. Consistent terminology reduces misunderstanding and prevents negative framing.

#### 5.3.6. Balanced Risk-Benefit Communication

Acknowledging potential risks while clearly explaining corresponding mitigation measures strengthens public credibility. Balanced communication is more effective than one-sided promotion, as it conveys integrity and promotes informed decision-making.

#### 5.3.7. Dialogue and Co-Creation

Dialogue and co-creation involve consumers, fishers, and community representatives early in pilot trials and public consultations to build mutual understanding and shared ownership of outcomes.

## 6. Framework and Reporting Checklist for Field Trials

The transition of GED fish from laboratory validation to commercial aquaculture settings requires a structured, transparent, and reproducible field trial framework. Many published studies still omit key methodological details, hindering cross-study comparability, biosafety evaluation, and regulatory review [30]. To address these gaps, a harmonized framework that integrates molecular, performance, welfare, and ecological endpoints is proposed. This approach would enhance scientific credibility, facilitate regulatory harmonization, and strengthen public confidence in GED aquaculture.

### 6.1. Rationale for a Structured Framework

Laboratory experiments cannot fully capture the complexity of real aquaculture systems, where water quality, microbial communities, pathogen loads, and environmental fluctuations interact dynamically [134]. Without a structured and standardized framework, trial outcomes remain fragmented, non-comparable, and often non-transparent. A well-designed and standardized reporting framework ensures completeness, reproducibility, and policy relevance, aligning scientific studies with biosafety and sustainability goals [24,70].

### 6.2. Core Components of the Framework

A robust framework for the field evaluation of GED fish requires the integration of five interdependent domains: molecular characterization, performance management, welfare monitoring, biosafety assessment, and transparent reporting. Each component ensures reproducibility, comparability, and credibility of results while aligning with regulatory and ethical expectations. The proposed structure builds on emerging recommendations from the OECD, FAO, and recent methodological syntheses in aquaculture biotechnology [29,30,69].

#### 6.2.1. Molecular Characterization

Accurate molecular validation supports all subsequent field analyses. Each GED fish line should be verified for the intended edits using deep sequencing, while potential off-target mutations must be screened across the genome using unbiased genome-wide approaches such as GUIDE-seq or CIRCLE-seq [135]. Heritable transmission of edits must be confirmed from founders (F0) to successive generations (F1 and F2) to ensure heritability and phenotypic stability [134]. For regulatory classification under non-transgenic categories, the absence of exogenous DNA or vector sequences must be demonstrated [18,99]. Clear reporting of the edit type (e.g., knockout, base substitution, or small indel change) and allele state (homozygous, heterozygous, or mosaic) enhances reproducibility and facilitates future meta-analyses [29,30].

#### 6.2.2. Performance Evaluation

Performance trials should assess growth, feed efficiency, survival, and reproduction under realistic aquaculture conditions. Genetically matched control lines must be maintained under identical environments to isolate the effects of genome editing. Long-term and multi-site trials across gradients of temperature, salinity, and dissolved oxygen can reveal genotype-by-environment interactions and trait robustness [70,126]. Standardized measurements of specific growth rate, feed conversion ratio, and reproductive output are essential for regulatory comparability.

#### 6.2.3. Welfare and Health Monitoring

Animal welfare is integral to the ethical and scientific integrity of GED fish trials. Welfare assessment should encompass physiological, behavioral, and morphological indicators of health. Physiological parameters such as cortisol concentration, heat shock protein expression, oxidative stress markers, and immune gene activation provide quantitative indicators of stress [136,137]. Behavioral endpoints, including swimming activity, aggression, and feeding frequency, should complement physiological data [138]. Morphological integrity can be assessed through radiographic or morphometric analyses to detect skeletal deformities, tissue damage, or fin erosion [139]. All procedures should comply with institutional animal-care guidelines and ARRIVE 2.0 principles [140].

#### 6.2.4. Biosafety and Ecological Containment

Biosafety monitoring addresses potential environmental interactions and escape risks of GED fish. Trials should quantify the likelihood of escape, post-escape survival, and potential ecological interactions such as competition, predation, and hybridization [29]. Controlled mesocosm or semi-natural experiments are suitable for testing under regulated exposure conditions [141,142]. To prevent gene flow, containment strategies can be enhanced through biological confinement (e.g., *dnd*-knockout sterility or hormonal control) [7]. Molecular tagging using SNP barcoding or CRISPR-traceable markers can facilitate identification of escaped individuals [143]. Environmental DNA (eDNA) surveillance can provide early detection of escapees and supports adaptive management [144].

#### 6.2.5. Data Transparency and Reporting

Adherence to the FAIR (Findable, Accessible, Interoperable, Reusable) data principles ensures that trial outputs contribute to a cumulative and verifiable evidence base. All genomic, phenotypic, and environmental datasets should be deposited in open repositories such as the NCBI Sequence Read Archive or Dryad [145]. Publications should include detailed protocols, statistical workflows, and negative or null findings to reduce bias [69]. Adopting standardized templates modeled on ARRIVE 2.0 guidelines facilitates regulatory evaluation and inter-institutional comparability [140]. Integrating socio-economic indicators such as production cost, market response, and consumer perception situates technical findings within a broader sustainability framework, thereby linking scientific rigor with societal accountability [29,70]. Through the systematic application of these components, field evaluations of GED fish can achieve scientific transparency, ethical compliance, regulatory credibility, and public trust. However, several technical bottlenecks remain before large-scale commercial adoption is feasible. Species with long generation cycles considerably slow the verification of edit stability, phenotypic penetrance, and transgenerational ecological fitness, making full life-cycle evaluations difficult within realistic research timelines [21,97]. In addition, excluding low-frequency off-target edits across large and repetitive aquaculture genomes also remains technically challenging, as genome-wide unbiased screening and deep-coverage sequencing are not yet routinely feasible at commercial scale [146]. Mosaicism in founder (F0) fish can further complicate trait prediction across generations, underlining the need for standardized allele-resolution protocols [147,148]. Additionally, full-duration field trials for large-bodied commercial species impose high financial and logistical costs, creating barriers for both industry and regulators and slowing progress toward globally harmonized validation pipelines [97]. Addressing these constraints will be essential for ensuring scientific robustness and regulatory confidence as GED fish move toward wider commercialization.

### 6.3. Proposed Reporting Checklist

To promote methodological consistency and transparency across studies, Table 5 presents a structured reporting checklist summarizing the minimum methodological, welfare, biosafety, and data transparency requirements recommended for the field evaluation of GED fish. Adoption of such a checklist enables cross-study comparability, supports regulatory review, and encourages community-level data synthesis and meta-analysis.

## 7. Conclusions and Future Perspectives

Genome editing has reshaped aquaculture biotechnology by providing precise, efficient, and cost-effective breeding tools to enhance traits primarily aimed at improving productivity, while potential contributions to welfare and sustainability remain hypothetical and require empirical demonstration. Over the past decade, the feasibility of CRISPR/Cas9-mediated genome editing has been firmly demonstrated in a wide range of aquaculture species, including Nile tilapia, red seabream, flounder, tiger puffer, salmonids, catfish, and carps [3,7,10,11,12,13]. Laboratory achievements are now beginning to translate into real-world applications, exemplified by the commercial release of GED red seabream, tiger puffer, and olive flounder in Japan and *mstn*-edited Nile tilapia in Argentina and Brazil [29,84]. These pioneering cases indicate that GED fish are no longer theoretical possibilities but an emerging component of global aquaculture. However, their long-term success will depend on transparent governance, standardized field evaluation, and sustained public engagement.

Despite significant progress, the field remains uneven in maturity and scope. Most GED lines have only been tested under controlled or semi-controlled conditions, and large-scale, multi-site field validation remains rare [134]. The absence of harmonized reporting standards complicates cross-study synthesis, while inconsistent welfare and biosafety monitoring hinders comparability. As identified in this review, structured field frameworks that integrate molecular characterization, performance, welfare, and ecological endpoints are urgently needed to enable transparent cross-country evaluation [30]. Future research should also incorporate socio-economic and consumer dimensions, linking technical efficiency with market acceptance and ethical responsibility. Demonstrating measurable benefits, such as reduced feed costs, lower disease incidence, or improved resource efficiency under commercial conditions, will be decisive for long-term viability.

From a governance perspective, the global regulatory landscape is evolving toward a clearer differentiation between transgenic and non-transgenic GED organisms. The case-by-case assessment principle adopted in Japan, Argentina, and Brazil is emerging as the most pragmatic regulatory model, as it aligns oversight stringency with the degree of genetic alteration and associated risk [70,125]. In contrast, regions that retain process-based classification, such as the EU, continue to face innovation bottlenecks that limit real-world testing. Over the next decade, policy convergence may occur through the development of internationally recognized frameworks for risk assessment, labeling, and traceability, coordinated by organizations such as the Codex Alimentarius Commission and the OECD working group on Biotechnology. Establishing common definitions, transparency requirements, and reporting templates will be critical to ensuring both regulatory efficiency and consumer confidence in global trade.

Ethical and societal dimensions will remain central to the sustainability of GED aquaculture. As public understanding of genome editing evolves, consumer acceptance will depend on clear and credible communication of its purpose and outcomes. Studies consistently show that transparency regarding editing methods, the absence of foreign DNA, and demonstrable environmental or welfare benefits enhances trust [86,149]. Future outreach should therefore move beyond risk explanation toward participatory communication and co-design involving consumers, producers, and regulators. Integrating ethical reflection into the early stages of research and commercialization can help prevent polarization and align technological development with social expectations.

The coming decade will likely witness expansion from proof-of-concept edits toward multi-trait optimization, combining improvements in growth, disease resistance, stress tolerance, and reproductive control. Advances in multiplex CRISPR, base editing, and prime editing are expected to further increase precision while minimizing unintended mutations. Integration of omics technologies, digital phenotyping, and machine learning may accelerate the discovery of gene–trait linkages and enable adaptive management of edited lines in variable environments [30]. At the same time, comprehensive ecological modeling and environmental DNA (eDNA) surveillance will be essential to assess potential ecosystem-level effects, ensuring that innovation proceeds responsibly.

In conclusion, genome editing offers aquaculture an unprecedented opportunity to balance productivity with sustainability, provided that its deployment is accompanied by rigorous evaluation, transparent regulation, and public accountability. Building on lessons learned from Japan and South America, the establishment of a globally harmonized framework that combines scientific rigor, ethical oversight, and data openness will be essential to realize the full potential of GED fish. Ultimately, the success of this technology will depend not only on genetic precision but on the collective ability of scientists, regulators, industry, and society to apply it wisely, equitably, and transparently.

## Figures and Tables

**Figure 1 cimb-47-01013-f001:**
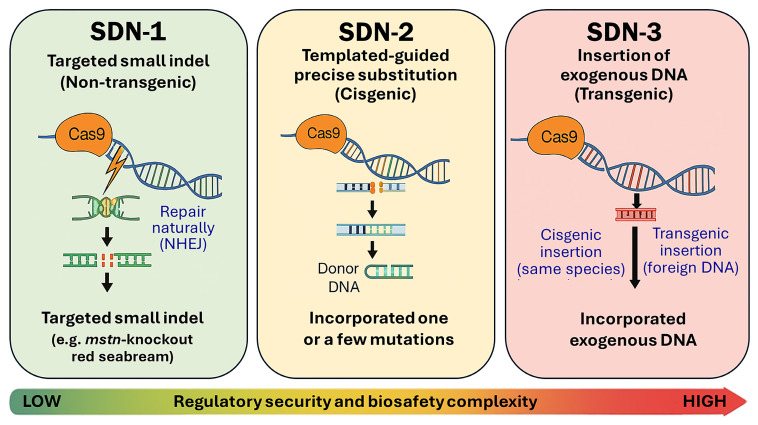
Overview of site-directed nuclease (SDN) classification in genome editing. SDN-1 produces small insertions or deletions (indels) through NHEJ without introducing foreign DNA; SDN-2 introduces precise substitutions using a short donor template; and SDN-3 inserts exogenous DNA, either cisgenic (from the same species) or transgenic (from a different species). The horizontal scale illustrates the increasing gradient of regulatory scrutiny and biosafety complexity from SDN-1 (low) to SDN-3 (high). Adapted from Matsuo and Tachikawa, 2022 [18].

**Figure 2 cimb-47-01013-f002:**
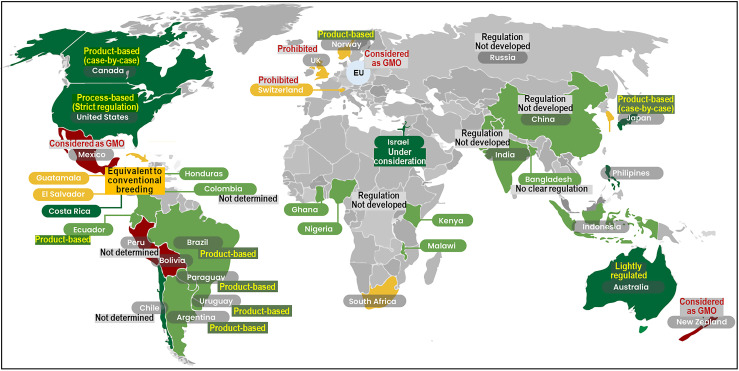
Global regulatory landscape for GED organisms as of 2025. Countries are categorized according to their regulatory approaches using the following color codes: no restriction (dark green); approved on a case-by-case basis (light green); debate ongoing or likely approval (yellow); prohibited or restricted (red); and undefined or developing frameworks or lacking formal policies (gray). Product-based systems such as those in Japan, Argentina, and Brazil (green) have enabled the early commercialization of GED fish, while the EU and New Zealand maintain GMO-equivalent regulation. (retrieved from the Global Gene Editing Regulation Tracker, Genetic Literacy Project; https://crispr-gene-editing-regs-tracker.geneticliteracyproject.org/, accessed on 29 November 2025).

**Figure 3 cimb-47-01013-f003:**
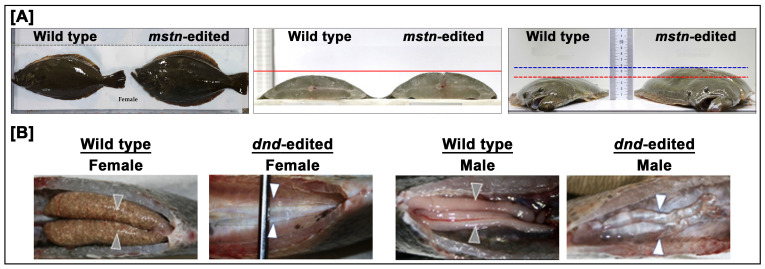
Comparative morphology of *mstn*-edited olive flounder and *dnd*-edited Nile tilapia. (**A**) *mstn*-edited olive flounder exhibit visibly increased body thickness and muscle mass relative to wild-type counterparts, confirming enhanced somatic growth following targeted knockout of the *mstn* gene. (**B**) Gonadal morphology of *dnd*-edited Nile tilapia shows complete germ cell depletion in both sexes. Wild-type female exhibit paired ovaries containing mature oocytes (arrowheads), while *dnd*-edited female lack discernible gonadal tissues. Similarly, wild-type males exhibit well-developed testes (arrowheads), while *dnd*-edited male show severe underdevelopment and sterility. These phenotypes collectively demonstrate the potential of CRISPR/Cas9-mediated editing of *mstn* and *dnd* genes to modulate muscle growth and reproductive control in aquaculture species. Images in panel (**A**) are adapted from Kim et al., 2019 [12] and Kim et al., 2024 [74], and images in panel (**B**) are adapted from Buchanan et al., 2021 [75].

**Figure 4 cimb-47-01013-f004:**
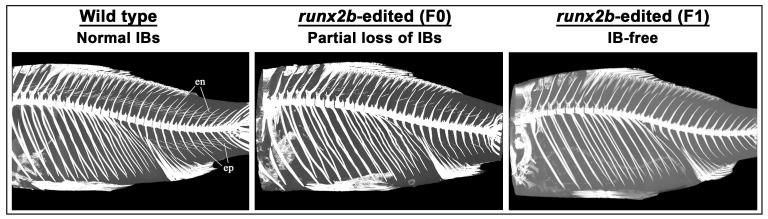
Micro-CT visualization of IB phenotypes in *runx2b*-edited Gibel carp. Representative micro-CT scans showing skeletal morphology of wild-type, *runx2b*-edited F0, and F1 individuals. In the wild type, numerous IBs are clearly visible between the epineural (en) and epipleural (ep) regions. The *runx2b*-edited F0 generation exhibited partial loss of IBs. In contrast, the *runx2b*-edited F1 progeny displayed a complete absence of IBs (IB-free phenotype), confirming stable germline transmission of the knockout. Images are adapted from Gan et al., 2023 [56].

**Figure 5 cimb-47-01013-f005:**
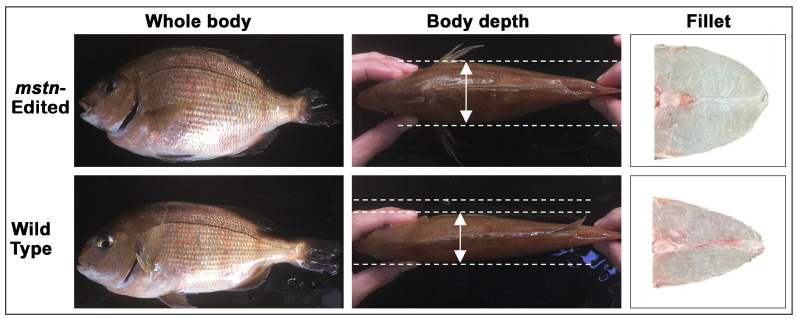
Comparative body morphology of *mstn*-edited and wild-type red seabream. Representative images show (from left to right) the whole body, dorsal view illustrating body depth, and transverse filet section. The *mstn*-edited fish display a noticeably deeper trunk and greater muscle mass compared with the wild type. Images are adapted from Kishimoto et al., 2018 [11] and the Japan Institute of Design Promotion (Good Design Award No. 9821; https://www.g-mark.org/gallery/winners/9821, accessed on 29 November 2025) [109].

**Figure 6 cimb-47-01013-f006:**
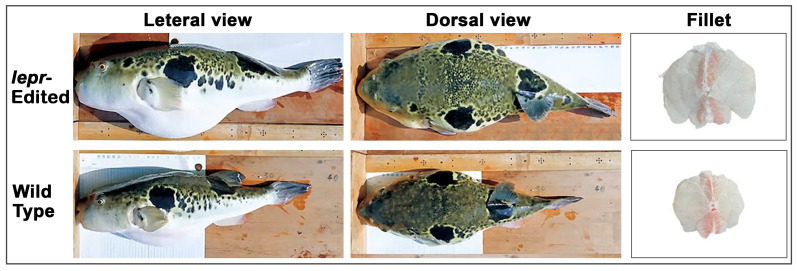
Morphological comparison between *lepr*-edited and wild-type tiger puffer. Representative images show lateral and dorsal views, as well as filet cross sections. The *lepr*-edited fish exhibited a rounder body shape and greater body mass compared with the wild type. Images adapted from Yasuda, 2023 [111] and the Japan Institute of Design Promotion [109].

**Table 1 cimb-47-01013-t001:** List of GED fish and shellfish species, target genes, and observed effects successfully achieved as of 2025. The list is arranged by year of publication, with only the first report for each species included.

SL	Species	Year	Target Gene(s)	Effect	Ref.
1.	Rainbow trout (*Oncorhynchus mykiss*)	2013	*sdY*	Induced male-to-female sex reversal	[8]
2.	Yellow catfish (*Pelteobagrus fulvidraco*)	2014	*mstnb*	Proof-of-concept study	[31]
3.	Nile tilapia (*Oreochromis niloticus*)	2014	*nanos*,* dmrt1*,* foxl2*	Produced sex reversal/sterility	[2]
4.	Atlantic salmon (*Salmo salar*)	2014	*tyr*,* slc45a2*	Induced loss of pigmentation	[32]
5.	Sea lamprey (*Petromyzon marinus*)	2015	*tyr*,* fgf8/17/18*	Induced loss of pigmentation	[33]
6.	Common carp (*Cyprinus carpio*)	2016	*sp7*,* mstn*	Increased muscle mass	[9]
7.	Northeast Chinese lamprey (*Lethenteron morii*)	2016	*gol*,* kctd10*	Induced loss of pigmentation	[34]
8.	Rohu carp (*Labeo rohita*)	2016	*tlr-22*	Decreased immunity	[35]
9.	Channel catfish (*Ictalurus punctatus*)	2016	*lh*	Produced sterile offspring	[36]
10.	Ridgetail white prawn (*Exopalaemon carinicauda*)	2016	*chi4*	Heritable editing	[37]
11.	Chinese tongue sole (*Cynoglossus semilaevis*)	2017	*dmrt1*	Disrupted spermatogenesis	[38]
12.	Red seabream (*Pagrus major*)	2018	*mstn*	Increased muscle mass	[11]
13.	Grass carp (*Ctenopharyngodon idella*)	2018	*jam-a*	Reduced viral replication	[39]
14.	Mexican tetra (*Astyanax mexicanus*)	2018	*oca2*	Produced albino offspring	[40]
15.	Olive flounder (*Paralichthys olivaceus*)	2019	*mstn*	Increased growth	[12]
16.	Tiger puffer (*Takifugu rubripes*)	2019	*mstn*	Improved growth by 1.95-fold	[13]
17.	Sterlet (*Acipenser ruthenus*)	2019	*dnd1*	Reduced primordial germ cells	[41]
18.	Pacific oyster (*Crassostrea gigas*)	2019	*mstn*	Induced muscle dysfunction	[42]
19.	Large-scale loach (*Paramisgurnus dabryanus*)	2019	*tyr*	Induced loss of pigmentation	[43]
20.	White crucian carp (*Carassius auratus cuvieri*)	2019	*tyr*	Induced loss of pigmentation	[44]
21.	Pacific bluefin tuna (*Thunnus orientalis*)	2019	*RyR1b*	Slow swimming behavior	[45]
22.	Blunt snout bream (*Megalobrama amblycephala*)	2020	*mstna*,* mstnb*	Improved growth	[46]
23.	Fathead minnow (*Pimephales promelas*)	2020	*tyr*	Induced loss of pigmentation	[47]
24.	Loach (*Misgurnus anguillicaudatus*)	2021	*mstn*	Increased growth	[48]
25.	Mackerel tuna (*Euthynnus affinis*)	2021	*slc24a5*	Reduced melanin pigments	[49]
26.	Large yellow croaker (*Larimichthys crocea*)	2022	*mstn*	Produced F0 mutant founders	[50]
27.	Malawi cichlid (*Astatotilapia calliptera*)	2022	*oca2*	Reduced melanin production	[51]
28.	Chub mackerel (*Scomber japonicus*)	2022	*slc45a2*	Produced albino offspring	[52]
29.	Freshwater prawn (*Macrobrachium rosenbergii*)	2022	*pax6*	Developed distinct eye phenotype	[53]
30.	Blotched snakehead (*Channa maculata*)	2023	*mstn*	Produced F0 mutant founders	[54]
31.	Crucian carp (*Carassius auratus*)	2023	*bmp6*	Eliminated IBs in F2 mutants	[55]
32.	Gibel carp (*Carassius gibelio*)	2023	*runx2b*	Eliminated IBs in F3 mutants	[56]
33.	African killifish (*Nothobranchius furzeri*)	2023	*mitfa*,* ltk*,* csf1ra*	Generated transparent fish	[57]
34.	Topmouth culter (*Culter alburnus*)	2024	*mstn*	Increased muscle growth	[58]
35.	Nibe croaker (*Nibea mitsukurii*)	2024	*dnd*	Eliminated germ cells	[59]
36.	Striped catfish (*Pangasianodon hypophthalmus*)	2024	*dnd1*	Reduced primordial germ cells	[60]
37.	Atlantic cod (*Gadus morhua*)	2024	*slc45a2*	Developed albino-like larvae	[61]
38.	Pacific abalone (*Haliotis discus hannai*)	2024	*β-tubulin*	Affected cilia development	[62]
39.	Large shoveljaw fish (*Onychostoma macrolepis*)	2025	*mstnb*,* tyr*,* bmp6*	Improved growth, reduced IBs	[63]
40.	Silver carp (*Hypophthalmichthys molitrix*)	2025	*bmp6*	Reduced IBs by 30% in F0 mutants	[64]
41.	Freshwater angelfish (*Pterophyllum scalare*)	2025	*dnd1*	Produced a male-biased sex ratio	[65]
42.	Largemouth bass (*Micropterus salmoides*)	2025	*tyrb*,* csf1ra*	*tyrb* mutation induced albinism	[66]

**Table 2 cimb-47-01013-t002:** Summary of GED fish lines evaluated under field or semi-field conditions, their main traits, system types, and identified research gaps.

Country/Study	Trait	Trial System Type	Control	Welfare/Health Metric	Generation Tracked	Data Access	Key Gaps
Japan: Red seabream (mstn-KO)	Growth	Land-based aquaculture facility	Wild-type/sibling line	Mortality, growth rate, feed use efficiency	F2	Government and regulatory reports (partial public data)	Limited ecological interaction data
Japan: Tiger puffer (*lepr*-KO)	Appetite, growth	Land-based aquaculture facility	Non-edited control	Growth, feed conversion, health screening	F1	Regulatory dossiers and government summaries (partial public data)	Limited multi-generation data
Japan: Olive flounder (*lepr*-KO)	Appetite, growth	Land-based aquaculture facility	Non-edited control	Growth, feed conversion, health screening	F1	Regulatory dossiers and government summaries (partial public data)	Limited multi-generation data
Brazil/Argentina: Nile tilapia (*mstn* KO)	Growth	Earthen pond	Conventional tilapia strains	Survival, growth, fertility checks	F1/F2	Company and regulatory documents, press release	Lack of peer-reviewed, open data

**Table 3 cimb-47-01013-t003:** GED fish lines advancing towards commercial production, as of 2025.

Species	Brand and Strain Name	Country	Institute	Year	Edited Gene	Trait
Red seabream (*Pagrus major*)	22nd century red seabream (4D strain)	Japan	Regional Fish Institute Ltd., Kyoto University, Kindai University	2021	*mstn*	Growth
Tiger puffer (*Takifugu rubripes*)	22nd century fugu (13D strain)	Japan	Regional Fish Institute Ltd., Kyoto University	2021	*lepr*	Appetite, growth
Olive flounder (*Paralichthys olivaceus*)	22nd century flounder (8D strain)	Japan	Regional Fish Institute Ltd., Kyoto University	2023	*lepr*	Appetite, growth
Nile tilapia (*Oreochromis niloticus*)	FLT-01	Argentina	AquaBounty	2018	*mstn*	Growth
	-	Brazil	CAT, Brazilian Fish	2025	*mstn*	Growth

**Table 4 cimb-47-01013-t004:** Comparative summary of regulatory frameworks for GED fish (as of 2025).

Country/Region	Regulatory Model	Legal Basis/Key Agencies	Scope and Criteria	Labeling Policy	Commercial Status
Japan	Product-based (SDN-1 exemption)	MHLW, MAFF, MOE (2019 framework)	Organisms without foreign DNA are considered outside GMO law; developer notification required	Mandatory labeling: QR code-based traceability	Three fish species approved (*mstn*-edited red seabream, *lepr*-edited tiger puffer, *lepr*-edited olive flounder)
Argentina	Product-based	Resolution No. 173/2015 (CONABIA)	Organisms are considered non-GMO in the absence of foreign DNA	Not required if classified as non-GMO	*mstn*-edited Nile tilapia authorized (2021)
Brazil	Product-based	CTNBio Normative Resolution No. 16/2018	Case-by-case determination of non-GMO status	Not required if classified as non-GMO	Pilot production of *mstn*-edited tilapia (2025)
USA	Product-based, risk-tiered	USDA and FDA (2020 modernization framework)	USDA exempts low-risk edits; FDA retains oversight for food safety and animal health	Voluntary disclosure	Research-stage GED fish; no commercial release
China	Developing hybrid model	MARA (2022 Guidelines)	Case-by-case biosafety evaluation and semi-field testing	To be determined	Research trials (carp, catfish); no market approvals yet
Canada	Process-neutral/precautionary	Health Canada (2021 Guidance)	All organisms with novel traits require premarket assessment	Case-by-case	No commercial approvals to date
European Union	Process-based (GMO Directive)	Directive 2001/18/EC and ECJ ruling (2018); EC, EFSA	All GED organisms regulated as GMOs	Mandatory GMO labeling	Field testing and commercialization are possible, but require full GMO authorization under Directive 2001/18/EC and Regulation (EC) No. 1829/2003

**Table 5 cimb-47-01013-t005:** Structured reporting checklist outlining minimum methodological, welfare, biosafety, and transparency requirements for field evaluation of GED fish.

Category	Reporting Elements	Minimum Requirement/Indicator	Rationale
Molecular characterization	Target gene, edit type, and verification method	Specify locus, CRISPR target, and edit category	Ensures edit traceability
	Verification of on-target edit and allele state	Deep sequencing; confirm homozygosity, heterozygosity, or mosaicism	Confirms molecular integrity
	Screening for off-target mutations	Apply unbiased methods (GUIDE-seq, CIRCLE-seq, WGS)	Demonstrates genomic precision
	Confirmation of absence of exogenous DNA	Use PCR or WGS to detect foreign fragments	Determines SDN-1 classification
	Validation of heritable transmission	Multi-generational verification (F0–F2)	Confirms germline stability
Performance evaluation	Quantification of production metrics	Report growth rate, feed conversion ratio, survival, and reproduction	Determines aquaculture viability
	Inclusion of matched control group	Parallel rearing under identical environmental conditions	Isolates genomic effects
	Multi-environment trial design	At least two sites or seasons	Tests robustness and scalability
Welfare and health	Assessment of physiological stress	Measure cortisol, HSPs, oxidative stress, and immune responses	Evaluates health resilience
	Behavioral monitoring	Record activity, aggression, feeding rate, and social interaction	Detects welfare compromise
	Morphological integrity	Conduct radiographic or deformity analysis; document mortality	Identifies unintended effects
	Ethics and oversight	Institutional animal care approval and ARRIVE 2.0 compliance	Ensures ethical compliance
Biosafety and ecological	Escape probability and containment assessment	Reports on physical and biological confinement measures	Prevents environmental release
	Simulated ecological exposure	Use mesocosm or semi-natural setup experimental setups	Evaluates ecological risk
	Biological confinement	Apply sterility strategies (e.g., *dnd*-knockout, triploidy, hormonal induction)	Minimizes gene flow and reproductive risk
	Environmental monitoring	Conduct eDNA surveillance	Enables early detection of escapees
	Apply molecular or phenotypic tagging	Apply SNP barcoding or CRISPR-based markers	Enables traceability
Data Transparency and reporting	Deposition of raw data and metadata	Submit to NCBI SRA, Dryad; ensure FAIR compliance	Enables data reuse
	Reporting of statistical methods and null results	Include model details and all outcomes in publication	Reduces reporting bias
	Inclusion of socio-economic metrics	Record production cost, market response, and perception metrics	Links science to sustainability

## Data Availability

No new data were created or analyzed in this study. Data sharing is not applicable to this article.

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
