# Peer review of "Genome-Edited Fish in the Field"

_cimb, 2025, doi:10.3390/cimb47121013_

Round 1

Reviewer 1 Report

Comments and Suggestions for Authors

This manuscript provides a systematic summary of the advancements in the application of CRISPR in aquaculture. It included molecular mechanisms, trait improvement, field trials, regulatory policies, and public perception, demonstrating high applied value. The manuscript progresses logically from the introduction to the conclusion, with a well-organized structure. In particular, the clear explanation of the SDN classification (SDN-1/2/3) helps readers understand the regulatory and commercial distinctions among different types of edits. Overall, the article meets the publication standards, although there are still some areas that need revision.

The authors also detail multiple cases of GED fish that have been commercialized or entered field trials, such as red seabream, tiger puffer, and olive flounder in Japan, as well as Nile tilapia in Argentina and Brazil. These examples are supported by figures and tables, enhancing the persuasiveness of the arguments. However, on pages 11-14, there is noticeable repetition in the descriptions of regulatory processes and measures (3.1 red seabream, 3.2 tiger puffer, and 3.3 olive flounder). Each case independently and elaborately outlines similar steps under Japan’s product-oriented approval framework, including case-specific confirmations by MAFF and MHLW, conclusions from the Biodiversity Impact Assessment Committee, highly similar physical containment measures, and transparency-based voluntary labeling policies. It is recommended that the highly consistent content in the three cases be refined and integrated.

On pages 18-20, the author proposes several key factors influencing public acceptance, such as perceived naturalness and transparency, and communication strategies, offering a valuable conceptual framework. However, the arguments in this section rely heavily on summative statements. We suggest that the author provide some original survey reports or literature as evidence, which would significantly enhance the persuasiveness of the claim that increasing transparency improves consumer perception.

On Page 18, Line 703, the author has incorrectly used inconsistent citation formats, and it is recommended to make corrections.

On page 23, the depiction of future technological directions leans toward listing positive prospects, with insufficient exploration of existing specific technical bottlenecks,such as the practical challenges of assessing editing stability and excluding off-target effects in fish with long generation cycles.

Comments on the Quality of English Language

In terms of language and presentation, this paper might benefit from English language polishing to improve the readability .

Author Response

Response to Comments of Reviewer 1

Comment 1: The authors also detail multiple cases of GED fish that have been commercialized or entered field trials, such as red seabream, tiger puffer, and olive flounder in Japan, as well as Nile tilapia in Argentina and Brazil. These examples are supported by figures and tables, enhancing the persuasiveness of the arguments. However, on pages 11-14, there is noticeable repetition in the descriptions of regulatory processes and measures (3.1 red seabream, 3.2 tiger puffer, and 3.3 olive flounder). Each case independently and elaborately outlines similar steps under Japan’s product-oriented approval framework, including case-specific confirmations by MAFF and MHLW, conclusions from the Biodiversity Impact Assessment Committee, highly similar physical containment measures, and transparency-based voluntary labeling policies. It is recommended that the highly consistent content in the three cases be refined and integrated.

Response: We thank the reviewer for this constructive suggestion. We agree that the original Sections 3.1–3.3 contained repeated descriptions of Japan’s regulatory approval framework for SDN-1 GED fish. To improve clarity and reduce redundancy, we consolidated the regulatory content shared by all species into a single unified paragraph placed before Section 3.1. This paragraph summarizes the product-based SDN-1 pathway administered by MAFF and MHLW and applies to all currently commercialized GED fish lines in Japan.

Each species subsection (3.1.2 Red Seabream, 3.1.3 Tiger Puffer, and 3.1.4 Olive Flounder) has been revised to focus exclusively on species-specific features including the edited gene, mutation details, trait outcomes, performance data, genomic validation, multigenerational stability, and commercialization strategies. In addition, each subsection has been revised with relevant biological and production information to preserve scientific depth while avoiding repetition.

We believe this restructuring strengthens readability and provides a clearer and more informative presentation of the three commercialized GED fish cases without duplicating regulatory descriptions.

Comment 2: On pages 18-20, the author proposes several key factors influencing public acceptance, such as perceived naturalness and transparency, and communication strategies, offering a valuable conceptual framework. However, the arguments in this section rely heavily on summative statements. We suggest that the author provide some original survey reports or literature as evidence, which would significantly enhance the persuasiveness of the claim that increasing transparency improves consumer perception.

Response: Thank you for this thoughtful comment. We acknowledge that original version in this section relied mainly on conceptual discussion supported by institutional reports, with only limited incorporation of survey-based studies on consumer attitudes towards GED foods. In response, we have revised the subsection on consumer acceptance, by integrating additional empirical evidence from recent consumer studies demonstrating that labeling, traceability, and clear communication of the editing purpose and absence of foreign DNA increase acceptance of GED foods with references. We have also included findings showing that transparent disclosure reduces perceived risks and increases willingness to purchase GED seafood in both online and restaurant markets. These sentences were inserted in section “5.2.2. Transparency” immediately following the explanatory of transparency as a determinant of acceptance. We believe this revision strengthens the scientific basis of the discussion and directly addresses the reviewer’s concern.

Comment 3: On Page 18, Line 703, the author has incorrectly used inconsistent citation formats, and it is recommended to make corrections.

Response: Response: Thank you for your observation. The reference has been corrected to numerical format to maintain consistency with the journal citation style.

Comment 4: On page 23, the depiction of future technological directions leans toward listing positive prospects, with insufficient exploration of existing specific technical bottlenecks,such as the practical challenges of assessing editing stability and excluding off-target effects in fish with long generation cycles.

Response: We thank the reviewer for highlighting the need to balance future opportunities with current technical limitations. We agree that the original version of this section placed greater emphasis on positive technological prospects while insufficiently addressing existing bottlenecks that may hinder large-scale commercial adoption of GED fish. In response, we have revised Section 6.2.5 Data transparency and reporting by adding a detailed paragraph explicitly outlining major practical constraints, including (i) the prolonged generation cycles of many aquaculture species that slow full life-cycle validation of edit stability and phenotypic penetrance, (ii) the difficulty of detecting low-frequency off-target edits in large and repetitive fish genomes due to the absence of scalable genome-wide monitoring platforms, (iii) CRISPR-induced mosaicism in F0 founders that complicates trait prediction across generations, and (iv) the high financial and logistical burden of full-duration field trials for large-bodied species. We believe the revision provides a more realistic depiction of current technological limitations while preserving a forward-looking perspective.

Reviewer 2 Report

Comments and Suggestions for Authors

The authors provide an overview of genome-edited fish applications that have been developed worldwide as well as a comprehensive discussion of those GE fish applications that have reached the market. The review therefore provides a useful source of information for researchers and risk assessors. In addition, the authors review regulatory frameworks for GE fish at a global scale. They call for a common framework for reporting field trial results based on the recognition that many assessments are restricted to contained conditions during product development, or reports are either fragmentary or are not disclosed, thereby providing a useful and valuable addition to the current discussions on risk  assessment and sustainability assessment of GE animals in general, and specifically GE fish.

Abstract:

line 17 ff: The authors state that market-approved SDN1 fish lines have demonstrated improved productivity without adverse welfare effects. While the improved productivity has been demostrated, there is – to my knowledge - no evidence for the absence of adverse welfare effects for these fish lines. It remains unclear whether welfare assessments were part of the regulatory assessments of these GED fish in the respective countries where these fish have gained market approval. As evidenced for growth-enhanced salmon, growth enhancement trait can have detrimental effects on physiological, developmental, or behavioral traits (see Devlin et al. 2015, 2020 and Li et al. 2009).

Devlin, R.H.; Leggatt, R.A.; Benfey, T.J. Genetic modification of growth in fish species used in aquaculture: Phenotypic and physiological responses. Fish Physiol. 2020, 38, 237–272; ISBN 9780128207949

Devlin, R.H.; Sundström, L.F.; Leggatt, R.A. Assessing Ecological and Evolutionary Consequences of Growth-Accelerated Genetically Engineered Fishes. BioScience 2015, 65, 685–700. https://doi.org/10.1093/biosci/biv068.

Li, D.; Hu, W.; Wang, Y.; Zhu, Z.; Fu, C. Reduced swimming abilities in fast-growing transgenic common carp Cyprinus carpio associated with their morphological variations. J. Fish Biol. 2009, 74, 186–197. https://doi.org/10.1111/j.1095-8649.2008.02128.x

Introduction

Line 102 ff

The authors refer to the regulations in the European Union. Line 105 states that “ …any potential field or containment trail involving GED fish must undergo full GMO authorization, leading to very limited or no field trials to date”. However, this is not fully correct, as experimental releases of GMOs into the enviornment, also known as Part B field trials according to Directive 2001/18/EC underly less stringent authorization requirements than authorizations for commercial placing on the market of GMOs (acc. To Part C Dir 2001/18/EC or Regulation (EC) No. 1829/2003). In addition, it is not correct that no field trials are carried out with genome edited organisms in the EU. The GMO register of the European Commission  (https://ec.europa.eu/food/food-feed-portal/screen/gmob/search) lists all experimental releases of GMOs including genome edited organisms. Please change accordingly.

Figure 2

The map of the global regulatory landscape for GED organisms gives an unclear picture of the provisions in the European Union. GEDs are considered GMOs in the EU and their cultivation or use is not prohibited but must undergo an authorization procedure with case-by-case risk assessment requirements (for details see Eckerstorfer et al. 2019, https://doi.org.10.3389/fbioe.2019.00026). Please amend the map accordingly.

In general, the readability of the figure could be improved.

Line 235:

Please write out abbreviations upon first use – IBs

Line 284 ff:

The authors conclude that the examples of trait categories modified in GED fish illustrate the aim to improve, among other, animal welfare. However, all trait categories reported aim to optimize productivity in addition to achieving better consumer acceptability (i.e. product quality). While disease resistance may be viewed as fostering animal welfare it can also be viewed as improving endurance of adverse production conditions and exacerbating negative states of industrialized food production (for discussion see also Dolezel et al. 2025, https://doi.org/10.3390/ani15182731). It is therefore debateable to view traits that target reduction of skeletal structures as welfare-related. Evidence for absence of adverse effects of these traits on fish welfare, e.g. swimming abilities or well-being, is currently lacking. Otherwise please indicate relevant scientific literature.

Line 382 ff: Examples of GED fish in the field are extensively described. However, the underlying assessment reports that evidence the absence of adverse effects are issued by the respective National (e.g. Japanese) regulatory authorities but no published accessible evidence is available for risk assessors. For example, it remains unclear whether health and welfare aspects have been assessed and the impression remains that the focus of most assessments was on food safety. It would have been valuable to have a comparison (e.g., in a table) of which assessments have been made for each GE fish application and whether these differ.

Lines  600 and 611 ff: Regulatory triggers differ between the EU and Canada, e.g. product-oriented regulatory triggers are used in Canada (e.g. trait novelty), while some authors consider triggers in the EU as a combination of process and product-oriented (see Eckerstorfer et al. 2019 https://doi.org.10.3389/fbioe.2019.00026). 

Line 626: the EU regulatory proposal refers only to certain new genomic techniques and only to plants! GED animals will remain to be regulated as GMOs.

Table 4 last line/last column: in the EU, field testing and commercialisation/market approval of GED fish is possible requiring application and authorization according to regulatory requirements. Please specify.

Line 703: references should be numbers

Line 867 ff: So far, and as reported for the examples of growth-enhanced GED fish discussed by the authors, the enhanced traits in GE fish currently aim to improve breeding efficiency and productivity of aquaculture. Therefore the statement that animal welfare and sustainability are also improved is highly speculative and requires still to be demonstrated. As indicated above (see abstract), a range of welfare implications of growth-enhanced fish have been documented in scientific literature. Please restate the sentence.

Line 918: should read DNA instead of DAN

Author Response

Response to Comments of Reviewer 2

Comment 1: [line 17 ff] The authors state that market-approved SDN1 fish lines have demonstrated improved productivity without adverse welfare effects. While the improved productivity has been demonstrated, there is – to my knowledge - no evidence for the absence of adverse welfare effects for these fish lines. It remains unclear whether welfare assessments were part of the regulatory assessments of these GED fish in the respective countries where these fish have gained market approval. As evidenced for growth-enhanced salmon, growth enhancement trait can have detrimental effects on physiological, developmental, or behavioral traits (see Devlin et al. 2015, 2020 and Li et al. 2009).

Response: Thank you for this valuable comment. We agree that, although improved productivity of market-approved SDN-1 fish has been reported, publicly available peer-reviewed evidence confirming the absence of adverse welfare is currently limited. We also acknowledge that it is unclear whether formal welfare metrics were included in all regulatory assessments, and that growth-enhancing traits can sometimes lead to physiological or behavioral trade-offs, as shown in growth-enhanced salmon. Accordingly, we revised the abstract to adopt a more cautious wording. The phrase “without adverse welfare effects” has been replaced with “

Comment 2: [Line 102 ff] The authors refer to the regulations in the European Union. Line 105 states that “ …any potential field or containment trail involving GED fish must undergo full GMO authorization, leading to very limited or no field trials to date”. However, this is not fully correct, as experimental releases of GMOs into the environment, also known as Part B field trials according to Directive 2001/18/EC underly less stringent authorization requirements than authorizations for commercial placing on the market of GMOs (acc. To Part C Dir 2001/18/EC or Regulation (EC) No. 1829/2003). In addition, it is not correct that no field trials are carried out with genome edited organisms in the EU. The GMO register of the European Commission  (https://ec.europa.eu/food/food-feed-portal/screen/gmob/search) lists all experimental releases of GMOs including genome edited organisms. Please change accordingly.

Response: We thank the reviewer for this important clarification. We agreed that our sentence did not sufficiently distinguish between commercial authorization (Part C of Directive 2001/18/EC/Regulation (EC) No. 1829/2003) and experimental environmental release (Part B of Directive 2001/18/EC). We also acknowledge that the European Commission GMO register includes experimental releases of GED organisms and therefore field trials are not completely absent in the EU. To address this, we revised the sentence in the introduction to read:

“Because GED organisms fall under the same legal framework as GMOs in the EU, commercial authorization is highly stringent, and although experimental releases are possible under Part B authorization, field trials remain limited compared to countries operating product-based SDN-1 frameworks.”

Comment 3: [Figure 2] The map of the global regulatory landscape for GED organisms gives an unclear picture of the provisions in the European Union. GEDs are considered GMOs in the EU and their cultivation or use is not prohibited but must undergo an authorization procedure with case-by-case risk assessment requirements (for details see Eckerstorfer et al. 2019, https://doi.org.10.3389/fbioe.2019.00026). Please amend the map accordingly. In general, the readability of the figure could be improved.

Response: We thank the reviewer for the helpful clarification regarding the regulatory status of GED organisms in the EU. We agree that our original coding for the EU implied prohibition rather than authorization under the GMO legal framework. In accordance with Directive 2001/18/EC and Regulation (EC) No. 1829/2003, and as outlined in by ckerstorfer et al. (2019), GED organisms are not prohibited in the EU but required authorization with case-by-case risk assessment. Accordingly, we have revised Figure 2 to classify the EU as “Regulated as GMOs” rather than “prohibited”. In addition, we have improved the readability of the map by adjusting contrast and enlarging labels.

Comment 4: [Line 235] Please write out abbreviations upon first use – IBs.

Response: Thank you for the comment. The abbreviation IBs (intermuscular bones) was already written out in full at its first appearance earlier in the manuscript (Line 178). Because journal guidelines require abbreviation expansion only at first use, and to avoid redundancy, we did not expand the term again at Line 235. We respectfully maintain the current format, as IBs has already been defined.

Comment 5: [Line 284 ff] The authors conclude that the examples of trait categories modified in GED fish illustrate the aim to improve, among other, animal welfare. However, all trait categories reported aim to optimize productivity in addition to achieving better consumer acceptability (i.e. product quality). While disease resistance may be viewed as fostering animal welfare it can also be viewed as improving endurance of adverse production conditions and exacerbating negative states of industrialized food production (for discussion see also Dolezel et al. 2025, https://doi.org/10.3390/ani15182731). It is therefore debateable to view traits that target reduction of skeletal structures as welfare-related. Evidence for absence of adverse effects of these traits on fish welfare, e.g. swimming abilities or well-being, is currently lacking. Otherwise please indicate relevant scientific literature.

Response: We thank the reviewer for this thoughtful comment. Our intention was not to imply that all trait categories currently explored in GED fish have been empirically demonstrated to improve welfare. We agree the most published traits primarily target productivity and product quality, and that evidence regarding direct welfare outcomes is still limited. To avoid overstating welfare implications, we have revised the text to indicate that some genome editing targets have been proposed as potential contributors to welfare (e.g. reduced aggression or reduced energy expenditure), but that empirical welfare assessment are still required. The sentence beginning with “Collectively, these examples illustrated…” has been updated to reflect that most current genome editing applications primarily target productivity and consumer appeal, while any potential welfare benefits remain hypothetical and require empirical validation. The revised sentence reads: “Collectively, these examples illustrate how genome editing in fish has progressed from exploratory mutagenesis to trait-driven innovation aimed at improving productivity and consumer appeal, with some traits also proposed as having potential welfare benefits; however, empirical welfare assessments remain limited.”

Comment 6: [Line 382 ff] Examples of GED fish in the field are extensively described. However, the underlying assessment reports that evidence the absence of adverse effects are issued by the respective National (e.g. Japanese) regulatory authorities but no published accessible evidence is available for risk assessors. For example, it remains unclear whether health and welfare aspects have been assessed and the impression remains that the focus of most assessments was on food safety. It would have been valuable to have a comparison (e.g., in a table) of which assessments have been made for each GE fish application and whether these differ.

Response: Thank you for this valuable comment. We agree that the regulatory decisions for currently commercialized GED fish rely on national assessment reports that are not publicly accessible, and therefore the empirical evidence underlying the absence of adverse effects cannot be independently reviewed by risk assessment. To avoid implying that such evidence is openly available or scientifically validated, we have revised the text to clarify that the regulatory conclusions in Japan, Argentina, and Brazil are based on internal evaluations by the respective authorities, and that transparency and accessibility of assessment data remain limited. We have also noted that publicly available information does not make it clear whether health or welfare aspects were evaluated beyond food safety considerations. At this stage, instead of adding a comparison table, we have added a sentence highlighting the need for harmonized reporting standards and publicly accessible assessments summaries across regulatory systems. The added sentence reads: “However, the extent and scope of these regulatory assessments remain difficult to evaluate because most underlying reports are not publicly accessible, and it is unclear whether systematic welfare or health evaluations were included beyond food safety considerations.”

Comment 7: [Lines  600 and 611 ff] Regulatory triggers differ between the EU and Canada, e.g. product-oriented regulatory triggers are used in Canada (e.g. trait novelty), while some authors consider triggers in the EU as a combination of process and product-oriented (see Eckerstorfer et al. 2019 https://doi.org.10.3389/fbioe.2019.00026). 

Response: Thank you for this helpful clarification. We agreed that while Canada applies a product-oriented trigger based on trait novelty, the EU trigger is not purely process-based. As discussed by Eckerstorfer et al. (2019), the EU regulatory system combines process- and product-oriented elements; the use of certain mutagenesis techniques determines inclusion under the GMO framework, after which a case-by-case assessment of the final product follows. We have revised the relevant sentences to reflect this combined process/product trigger rather than referring to the EU trigger as exclusively process-based.

Comment 8: Line 626: the EU regulatory proposal refers only to certain new genomic techniques and only to plants! GED animals will remain to be regulated as GMOs.

Response: Thank you for your correction. We agreed that the EU proposal on new genomic techniques applies exclusively to plants and does not extend to animals. To avoid misinterpretation, we have revised the sentence to clarify that GED animals will continue to be regulated as GMOs under Directive 2001/18/EC.  The revised sentence now reads: “However, the current EU regulatory proposal on new genomic techniques applies exclusively to plants; GED animals are not included in this reform and will continue to be regulated as GMOs under Directive 2001/18/EC.”

Comment 9: Table 4 last line/last column: in the EU, field testing and commercialisation/market approval of GED fish is possible requiring application and authorization according to regulatory requirements. Please specify.

Response: Thank you for this helpful comment. We agree that both field testing and commercialization of GED fish are possible in the EU, provided that applications obtain full GMO authorization under the existing legislative framework. To avoid implying prohibition, we have revised the EU entry in the last column of Table 4. The updated text now reads: “Field testing and com-mercialization are possi-ble, but require full GMO authorization under Directive 2001/18/EC and Regulation (EC) No. 1829/2003”

Comment 10: [Line 703] references should be numbers

Response: Thank you for your observation. The reference has been corrected to numerical format to maintain consistency with the journal citation style.

Comment 11: [Line 867 ff] So far, and as reported for the examples of growth-enhanced GED fish discussed by the authors, the enhanced traits in GE fish currently aim to improve breeding efficiency and productivity of aquaculture. Therefore the statement that animal welfare and sustainability are also improved is highly speculative and requires still to be demonstrated. As indicated above (see abstract), a range of welfare implications of growth-enhanced fish have been documented in scientific literature. Please restate the sentence.

Response: Thank you for this important comment. We agree that most growth-enhanced GED fish currently aim to improve breeding efficiency and productivity in aquaculture, and that welfare and sustainability outcomes have not yet been empirically demonstrated. To avoid these implications, we have revised the text to clarify that potential contributions to welfare and sustainability remain hypothetical and require validation. The original sentence has been replaced with following revised version: “Genome editing has reshaped aquaculture biotechnology by providing precise, efficient, and cost-effective breeding tools to enhance traits primarily aimed at improving productivity, while potential contributions to welfare and sustainability remain hypothetical and require empirical demonstration.”

Comment 12: Line 918: should read DNA instead of DAN

Response: Thank you for pointing this out. The typographical error has been corrected from “DAN” to “DNA.”
